# DUOX2-Induced Oxidative Stress Inhibits Intestinal Angiogenesis through MMP3 in a Low-Birth-Weight Piglet Model

**DOI:** 10.3390/antiox12101800

**Published:** 2023-09-25

**Authors:** Dongbin Zou, Yun Yang, Fengjie Ji, Renlong Lv, Tieshan Xu, Chengjun Hu

**Affiliations:** 1Tropical Crop Genetic Resource Research Institute, Chinese Academy of Tropical Agricultural Sciences, Haikou 571101, China; zdb0515a@126.com (D.Z.); yangyun2023@126.com (Y.Y.); fengjie_ji@126.com (F.J.); lvrenlong@aliyun.com (R.L.);; 2College of Life Sciences, Hainan University, Haikou 571101, China; 3College of Animal Science and Technology, Huazhong Agricultural University, Wuhan 430070, China

**Keywords:** DUOX2, intestinal angiogenesis, low birth weight, MMP3, oxidative stress

## Abstract

Intestinal vessels play a critical role in nutrient absorption, whereas the effect and mechanism of low birth weight (LBW) on its formation remain unclear. Here, twenty newborn piglets were assigned to the control (CON) group (1162 ± 98 g) and LBW group (724 ± 31 g) according to their birth weight. Results showed that the villus height and the activity of maltase in the jejunum were lower in the LBW group than in the CON group. LBW group exhibited a higher oxidative stress level and impaired mitochondrial function in the jejunum and was lower than the CON group in the intestinal vascular density. To investigate the role of oxidative stress in intestinal angiogenesis, H_2_O_2_ was employed to induce oxidative stress in porcine intestinal epithelial cells (IPEC-J2). The results showed that the conditioned media from IPEC-J2 with H_2_O_2_ treatment decreased the angiogenesis of porcine vascular endothelial cells (PVEC). Transcriptome analysis revealed that a higher expression level of dual oxidase 2 (DUOX2) was found in the intestine of LBW piglets. Knockdown of DUOX2 in IPEC-J2 increased the proliferation and decreased the oxidative stress level. In addition, conditioned media from IPEC-J2 with DUOX2-knockdown was demonstrated to promote the angiogenesis of PVEC. Mechanistically, the knockdown of DUOX2 decreased the reactive oxygen species (ROS) level, thus increasing the angiogenesis in a matrix metalloproteinase 3 (MMP3) dependent manner. Conclusively, our results indicated that DUOX2-induced oxidative stress inhibited intestinal angiogenesis through MMP3 in a LBW piglet model.

## 1. Introduction

It is estimated that about 15–20% of all newborns worldwide are considered as low birth weight (LBW) by the World Health Organization [1,2]. LBW is associated with fetus obesity, diabetes, and metabolic syndrome in humans [3], and impairs hippocampal neurogenesis and cognition in their offspring [4]. Moreover, LBW was demonstrated to reduce the rate of growth after birth [5,6]. Despite the strategy provided for improving the developing rate of LBW [7], the mechanism underlying the lower growth rate of LBW newborns remains largely unknown.

The small intestine serves as the organ for nutrient absorption. The study has demonstrated the reduced nutrient absorption in the small intestine with LBW [8], indicating that the impaired intestinal absorption function might contribute to a lower growth rate of LBW newborns. Intestinal blood vessels act as the channel for nutrients to the body, of which a poor formation could negatively affect the blood flow and nutrient supply. However, previous studies were mainly focused on how to improve the development of intestinal villus in LBW, neglecting the role of blood vessels in nutrient absorption [9,10]. Therefore, elucidating the influence and mechanism of the LBW on intestinal angiogenesis could contribute to comprehending its molecular bases. Oxidative stress plays a critical role in angiogenesis, which causes vascular endothelial cell injury at a high level, whereas stimulating angiogenesis at a low level [11]. Notably, LBW was demonstrated to be associated with elevated oxidative stress levels in the small intestine [12]. The evidence above indicated that LBW might reduce intestinal angiogenesis via increasing oxidative stress levels.

Considering the similar physiological characteristics to humans, pigs are often selected as a model to investigate the intestinal function of humans [8]. Therefore, the objectives of this study were to investigate the role and the underlying mechanism of LBW in intestinal angiogenesis using the model of LBW piglet.

## 2. Materials and Methods

### 2.1. Animals and Sample Collection

The sows were provided by the farm of Hainan Guangfu Agricultural Development Co., Ltd., Haikou, China. They were housed individually and turned to the farrowing rooms on day 110 of gestation. After parturition, the birth weight of each piglet was recorded to calculate the average birth weight. In this study, a total of 186 neonatal piglets were obtained, with an average birth weight of 1162 ± 98 g (means ± standard error). Twenty newborn piglets were selected and assigned to the normal birth weight group (CON group) and low birth weight group (LBW group). The average birth weight for the CON group and LBW group was 1162 ± 98 g and 724 ± 31 g (means ± standard error, *n* = 10), respectively.

On day 1 of birth, piglets were slaughtered. Approximately 2 cm segments of jejunal samples were cut and washed with ice-cold saline. The samples were fixed in 4% paraformaldehyde solution for histomorphology examination or immediately snap-frozen in liquid nitrogen and stored at −80 °C for further analysis.

### 2.2. Analysis of Jejunal Enzyme Activities

The obtained 0.5 g of jejunum sample was homogenized in cold saline and centrifuged at 3000× *g* and 4 °C for 10 min, with the supernatant collected, which was used to measure the activities of sucrase, maltase, and lactase. The enzyme activities were determined using the commercial kits (Jiancheng Bioengineering Ltd., Nanjing, China) according to the manufacturer’s instructions.

### 2.3. Oxidative Stress and Mitochondrial Biogenesis Parameters

The reactive oxygen species (ROS) levels in jejunum and cells were measured using 2′, 7′-dichlorofluorescein diacetates (DCFH-DA) as described previously [13]. In brief, cells were incubated with 200 μL DCFH-DA at 37 °C for 30 min and washed three times with phosphate buffered saline (PBS). The fluorescence intensity was measured using a fluorescent fluorometer with an excitation wavelength of 488 nm and an emission wavelength of 525 nm.

The activities of citrate synthase (CS), malondialdehyde (MDA), superoxide dismutase (SOD), glutathione peroxidase (GSH-Px), and glutathione (GSH) were determined using the commercial kits provided by Nanjing Jiancheng Bioengineering Institute (Nanjing, China). The activities of Complex I and Complex III were determined using commercial kits (Cominbio Co., Suzhou, China) according to the manufacturer’s instructions.

### 2.4. The Adenosine Triphosphate (ATP) Level in Jejunum

The level of ATP in the jejunum was determined using commercial kits (Beyotime Biotechnology, Jiangsu, China). Approximately 20 mg of jejunum was homogenized in 100 mL lysis buffer and centrifuged at 12,000× *g* and 4 °C for 10 min. The supernatant was analyzed according to the manufacturer’s instructions.

### 2.5. Mitochondrial DNA (mtDNA) Copy Number

Total genomic DNA was isolated from jejunum using the QIAamp DNA Mini Kit (Qiagen, Germantown, MD, USA) according to the manufacturer’s instructions. Mitochondrial DNA copy number was determined using primers for mitochondrial cytochrome b (Cytb) and acidic ribosomal phosphoprotein P0 (36B4) nuclear gene [14].

### 2.6. Real-Time Quantitative RT-PCR

Total RNA was isolated from the jejunum using the TRIzol reagent (Invitrogen, Carlsbad, CA, USA). The mRNA levels of target genes were determined as described previously [15]. The total reaction system contains 2 μL of cDNA template, 5 μL of RealStar Green Fast Mixture (GenStar, Beijing, China), 1.4 μL of water, and 0.8 μL of each primer. The thermal cycling consisted of a 10-min incubation at 95 °C, followed by 40 cycles of denaturation for 15 s at 95 °C and annealing/extension for 30 s at 60 °C. Relative gene expression was calculated using the formula 2^−ΔΔCT^ values. Primers are listed in Appendix A.

### 2.7. Western Blotting

Total protein was extracted from the jejunum using the RIPA buffer (Beyotime Biotechnology, Shanghai, China). The protein concentration was measured using a BCA protein assay kit (Beyotime Biotechnology, Shanghai, China). The protein was separated by SDS-PAGE gel and transferred onto the PVDF membrane as described previously [15]. The blots were incubated overnight at 4 °C adopting the following primary antibodies: Kelch-like ECH associated protein 1 (Keap1) (Proteintech, Rosemont, IL, USA), Occludin (Proteintech, Rosemont, IL, USA), Claudin 1 (Proteintech, Rosemont, IL, USA), vascular endothelial growth factor-A (VEGF-A) (Proteintech, Rosemont, IL, USA), platelet endothelial cell adhesion molecule-1 (CD31) (Abcam, Waltham, MA, USA), and β-Actin (CST, Danvers, MA, USA). The density of bands was quantified using Image J software 1.48v and normalized to β-Actin levels.

### 2.8. Immunohistochemistry Staining

Immunohistochemistry staining for activating transcription factor 6 (ATF6) was carried out as described previously [16]. ATF6 antibody (Servicebio, Wuhan, China) and peroxidase-conjugated goat anti-rabbit IgG (Servicebio, Wuhan, China) were adopted as the primary and secondary antibodies, respectively. Five photographs were selected randomly from each slide and analyzed using Image J software 1.48v.

### 2.9. Immunofluorescence Assay

The staining of jejunum was performed as described previously [16]. In brief, intestine tissues were paraffin-embedded and sectioned at a thickness of 5 μm, and then were deparaffinized in xylene and rehydrated in grade alcohol. Slides were blocked in 3% bovine serum albumin for 30 min, followed by incubation in CD31 primary antibodies (Abcam, Waltham, MA, USA) diluted at 4 °C overnight in a wet box. The slides were washed 5 times for 8 min in PBS and then incubated in CY3 rabbit anti-goat antibody (Servicebio, Wuhan, China) for 30 min. Slides were washed 3 times for 5 min and then stained with 4′, 6-diamidino-2-phenylindole solution (DAPI) for 10 min. Five photographs were selected randomly from each slide and analyzed using Image J software 1.48v.

### 2.10. Transcriptome Sequencing

Total RNA was isolated from the jejunum using the TRIzol reagent (Invitrogen, Carlsbad, CA, USA). RNA integrity was determined using 1% agarose gel electrophoresis. The total RNA was purified and reverse-transcribed into cDNA with SuperScript II Reverse Transcriptase (Invitrogen, USA). Sequencing was performed on an illumina Novaseq™ 6000 platform (LC-Bio Technology Co., Ltd., Hangzhou, China).

Fastp software (https://github.com/OpenGene/fastp, accessed on 10 July 2021) was used to eliminate adapters and low-quality reads. The mapped reads of each sample were assembled using StringTie (https://ccb.jhu.edu/software/stringtie, accessed on 10 July 2021). All transcriptomes from samples were merged to reconstruct a comprehensive transcriptome using GFFcompare (https://github.com/gpertea/gffcompare, accessed on 10 July 2021). StringTie was used to determine the expression levels of genes by calculating FPKM (Fragments Per Kilobase of exon model per Million mapped fragments). The differentially expressed genes (DEGs) were selected with fold change > 2 (or fold change < 0.5) and *p*-value < 0.05. DEGs were then subjected to the Gene Ontology (GO) enrichment analyses.

### 2.11. Cell Culture

The porcine intestinal epithelial cells (IPEC-J2) were cultured in Dulbecco’s modified Eagle’s medium (DMEM, Gibco, Thermo Fisher, Waltham, MA, USA) existing 10% fetal bovine serum (FBS; Gibco, Thermo Fisher, Waltham, MA, USA), 100 U/mL penicillin, and 100 μg/mL streptomycin, and maintained at 37 °C with 5% CO_2_. Porcine vascular endothelial cells (PVEC) were placed in 1640 medium for growing, with 10% FBS, 100 U/mL penicillin, and 100 μg/mL streptomycin at 37 °C in 5% CO_2_ atmosphere.

### 2.12. Collection of Conditioned Media

IPEC-J2 were seeded into a 6-well plate in a growth medium, and treated with 200 μM si-DUOX2 alone or in combination with MMP3 inhibitor UK356618 (MedChemExpress, Monmouth Junction, NJ, USA) for 48 h. After removing the growth medium, cells were cultured in serum-free medium for 24 h, which was centrifuged at 3000× *g* and 4 °C for 10 min and collected as conditioned media. conditioned media was stored at −80 °C before use.

### 2.13. Small Interfering RNA (siRNA) Transfection

PVEC were grown to 50% confluence in 6-well plates and transfected with Lipofectamine 2000 (Invitrogen, Carlsbad, CA, USA) as instructed by the manufacturer. The sequences were 5′-GCUUCUAGCUAUUAGUAAATT-3′ and 5′-UUUACUAAUAGCUAGAAGCTT-3′ for DUOX2 siRNA; 5′-UUCUCCGAACGUGUCACGUTT-3′ and 5′-ACGUGACACGUUCGGAGAATT-3′ for control siRNA. The siRNA was provided by GenePharma Co., Ltd. (Shanghai, China).

### 2.14. Tube Formation Assay

Briefly, 4 × 10^4^ PVEC in 100 μL conditioned media were seeded in 96-well plates pre-coated with 50 μL Matrigel (BD company, Franklin Lakes, NJ, USA). After incubation for 6 h, the images were captured with a microscope. The tube formation was analyzed using Image J software 1.48v.

### 2.15. Transwell Assay

Briefly, 4 × 10^4^ PVEC in 200 μL serum-free medium were added to the upper chamber, followed by 600 μL of conditioned media added to the lower chamber. After incubation for 48 h, cells were fixed with 4% paraformaldehyde solution and then stained with crystal violet solution. Images were captured with a microscope and analyzed using the Image J software 1.48v.

### 2.16. Wound Healing Assay

PVEC was seeded into 6-well plates. After incubated for 48 h, cells were wounded with a 10 μL pipette tip and maintained for 12 h in conditioned media. The wounded areas were captured using a microscope at 0 h and 12 h and analyzed using the Image J software 1.48v.

### 2.17. Statistical Analysis

Data are represented as mean ± SEM. Statistical analyses were performed using SPSS 20.0 (SPPS Inc., Chicago, IL, USA) software. The Student’s *t*-test was performed to analyze the differences between the two groups, with *p* < 0.05 considered statistically significant.

## 3. Results

### 3.1. The Morphology and Activities of Digestive Enzymes in Jejunum

LBW was reported to impair nutrient digestion and absorption, as well as the protein expression level of Occludin in the intestine of piglets [8,12]. Therefore, these indexes were evaluated to further confirm the successful establishment of the LBW model. The results showed that a decrease in villus height was observed in the LBW group in comparison to the CON group (Figure 1a–c), as well as the lower activities of sucrase and maltase (Figure 1d,e). However, no significant difference was observed in the activity of lactase between the two groups (Figure 1f). Additionally, the mRNA expression levels of amino acids transporters SLC1A5, SLC7A5, and SLC38A2 were lower in the LBW group in comparison to the CON group (Figure 1g), as well as the mRNA and protein expression levels of Occludin (Figure 1h,i). However, no difference was observed in the mRNA expression level of glucose transporters GLUT1, GLUT3, or GLUT4 between the two groups (Figure 1g). In addition, the protein expression level of Claudin 1 was lower in the LBW group in comparison to the CON group (Figure 1h,i). In general, the above-mentioned results showed that the LBW model was established successfully.

### 3.2. Oxidative Stress Level in Jejunum

Compared to the CON group, the LBW group displayed higher levels of ROS and protein carbonyl (Figure 2a,b), while a lower level of GSH in the jejunum (Figure 2c). Compared to the CON group, the LBW group has a tendency to increase the levels of MDA and 8-OHdG in the jejunum (Figure 2d,e). The two groups displayed no difference in the level of SOD or GSH-Px (Figure 2f,g). To further confirm that the intestines from LBW were subjected to higher oxidative stress, the mRNA expression levels of endoplasmic stress markers were determined. The results of qPCR and immunohistochemistry staining analysis indicated a significantly increased *ATF6* expression level in the LBW group in comparison to the CON group (Figure 2h–j). However, no significant difference was found in the mRNA expression level of ATF4, X-box binding protein 1 (XBP), glucose-regulated protein 78 (GRP78), or C/EBP-homologous protein (CHOP) between the two groups (Figure 2h). No difference was observed in the mRNA expression level of Keap 1 between the two groups (Figure 2k). However, the protein expression level of Keap 1 was higher in the LBW than in the CON group (Figure 2l,m). These results suggested that the intestines from LBW were subjected to higher oxidative stress.

### 3.3. ATP Level and Mitochondrial Function in Jejunum

The ATP level (Figure 3a) and CS activity (Figure 3b) were lower in the LBW group. However, no significant difference was found in the mtDNA content between the two groups (Figure 3c). Compared to the CON group, the LBW group displayed lower activities of Complex I and III (Figure 3d,e).

### 3.4. LBW Increased the Dual Oxidase 2 (DUOX2) Expression Level and Decreased Vessel Density in Jejunum

To reveal the mechanism of increased oxidative stress level in the LBW intestine, transcriptome analysis was conducted to investigate the DEGs in the jejunum between CON and LBW groups. Here, a total of 1140 DEGs were significantly changed in the jejunum, of which 742 were significantly up-regulated and 398 were significantly down-regulated (Figure 4a). The DEGs in jejunum between LBW and CON groups were distributed into vascular endothelial cell proliferation angiogenesis and reactive oxygen species metabolic process (Figure 4b,c). Compared with the CON group, the expression level of DUOX2 and dual oxidase maturation factor 2 (DUOXA2) in LBW was significantly up-regulated, whereas that of VEGF-A was significantly down-regulated (Figure 4d). In line with this result, the mRNA expression level of DUOX2 in the jejunum was higher in the LBW group than in the CON group (Figure 4e). No difference was observed in the gene expression level of other NADPH oxidases (e.g., NOX1, NOX3, NOX4, NOX5, DUOX1) in jejunum between the two groups (Appendix A). DUOX2, as part of a NADPH oxidase complex, is a major source of ROS in cells [17]. These results suggested that increased DUOX2 expression levels might have contributed to the increased oxidative stress level in the LBW intestine. Furthermore, the mRNA and protein expression level of CD31 was lower in the LBW group than in the CON group (Figure 4f–i). We also observed the lower mRNA and protein expression levels of VEGF-A in the LBW group in comparison to the CON group (Figure 4f,i). The results suggested that LBW impairs intestinal angiogenesis.

### 3.5. Oxidative Stress Disrupts Intestinal Angiogenesis

To investigate the role of oxidative stress in intestinal angiogenesis, H_2_O_2_ was employed to induce oxidative stress in IPEC-J2. As depicted in Figure 5, the treatment with 200 μM H_2_O_2_ resulted in decreased cell viability and increased ROS level in IPEC-J2 (Figure 5a–c). Oxidative stress induced by H_2_O_2_ decreased the protein expression level of VEGF-A in IPEC-J2 (Figure 5d,e). In addition, the conditioned media from H_2_O_2_-treated IPEC-J2 was found to decrease the migration (Figure 5f,g) and tube formation (Figure 5h,i) of PVEC. These results indicated that elevation of oxidative stress disrupts intestinal angiogenesis in vitro.

### 3.6. DUOX2 Regulates Intestinal Angiogenesis via Activating the MMP3 Pathway

To test the hypothesis that the increased DUOX2 expression level is related to poor angiogenesis in the intestine, we established DUOX2 knockdown expressing IPEC-J2 (Figure 6a). The results indicated that the knockdown of DUOX2 decreased the ROS level in IPEC-J2 (Figure 6b). The conditioned media from DUOX2-knockdown cells increased the PVEC tube formation (Figure 6c,d) and migration (Figure 6e,f). The differences in the mRNA expression levels of angiogenesis-associated factors in CON and DUOX2-knockdown cells were determined through qPCR (Figure 6g). The results showed that the knockdown of DUOX2 significantly increased the mRNA and protein levels of MMP3 (Figure 6g,h), as well as the proliferation of IPEC-J2 (Figure 6j), but has no effects on the mRNA stability of MMP3 (Figure 6i). UK356618, an inhibitor of MMP3, eliminated the promoting effects of DUOX2 silencing on PVEC migration (Figure 6k,l) and tube formation (Figure 6k,m) in vitro. These results suggested that DUOX2 regulates intestinal angiogenesis via activating the MMP3 pathway.

## 4. Discussion

LBW was indicated to be associated with impaired intestinal absorption function. In this study, we demonstrated that LBW contributed to the decreased villus height and protein expression level of Occludin and Claudin 1, accompanied by the decreased mRNA level of genes related to amino acid transporter in the intestine, which confirms the finding of previous studies [18,19]. Animal growth depends on nutrient supply, in a deficiency could lead to a lower growth rate. These results indicated that the postnatal growth restriction in LBW neonates might be attributed to the impaired intestinal absorption function. The intestinal stem cells serve to maintain the intestinal epithelial function and renewal [20]. Evidence has revealed the decreased positive cells and increased apoptosis in the intestine of LBW piglets [21]. Therefore, we suspected that the reduced villus height in LBW might be associated with the increased apoptosis of intestinal stem cells. Previous studies also demonstrated the compromised intestinal digestion function in the intrauterine growth retardation piglets [22]. In line with that, we found that the activities of maltase and sucrase in the intestine were significantly decreased in LBW piglets.

Oxidative stress acts as a critical factor for intestine development. The physiological levels of ROS serve as messengers and signaling molecules, whereas high levels of ROS are implicated in intestinal injury and pathological processes [23,24]. In the present study, we observed higher ROS production and lower GSH levels in the intestine for LBW piglets. High oxidative stress was reported to induce lipid peroxidation, DNA damage, and protein oxidation [25]. In line with the change of ROS levels in the intestine, the LBW piglets showed higher levels of MDA, 8-OHdG, and protein carbonyl in the intestine. Consistently, a previous study demonstrated the association of LBW with the increased levels of MDA and GSH in jejunum [12]. In addition, the LBW intestine also displayed an elevation in the mRNA and protein expression levels of endoplasmic reticulum stress markers ATF6. ROS can exacerbate protein misfolding in the endoplasmic reticulum lumen, thereby amplifying unfolding protein response signaling [26], while inhibition of ROS level by N-acetylcysteine could attenuate reticulum stress in cells [27]. Taken together, our results suggested that the intestine of LBW was subjected to greater oxidative damage.

ROS was derived from a variety of sources, including the mitochondrial respiratory chain, NADPH oxidase, xanthine oxidase, lipoxygenases, and myeloperoxidase [28], among which mitochondrial electron transport chain (I and III) serves as the major sites of ROS production [29]. In addition, mitochondrion is an attacking target for ROS, which induces mitochondrial dysfunction [30]. Therefore, the mitochondrial biogenesis in the intestine was investigated. In the present study, the mitochondrial biogenesis in the intestine was reduced in the LBW piglets, evidenced by the decreased ATP and CS levels, which was consistent with the published study [31]. The electron leakage from mitochondrial electron transport chain Complexes I and III contributes to the production of O^2−^ [28]. Therefore, the activities of subunits encoding the complexes of the electron transport chain were investigated. We found the decreased activities of mitochondrial Complex I and III in the intestine of the LBW piglets in comparison to the CON piglets, suggesting that decreased activities of mitochondrial Complex I and III might contributed to a higher ROS level in the intestine of LBW.

Intestinal blood vessels play an important role in nutrition absorption. However, the mechanism of LBW regarding the regulation of intestinal angiogenesis remains unclear. CD31 acts as a biomarker of vascular endothelial cells in blood vessels [30]. Here, the reduced immunofluorescence staining and protein level of CD31 in the intestine indicated the decreased intestinal vessel density in the LBW piglets when compared to the CON piglets. VEGF-A is a crucial angiogenic factor [32]. The LBW group was lower than the CON group in the protein level of VEGF-A, further indicating that LBW resulted in impaired intestinal angiogenesis. Previous study has demonstrated that oxidative stress could impair angiogenesis by inducing dysfunction of vascular endothelial cells [33]. Our previous study also illustrated that NADPH oxidase 2 impairs angiogenesis in the placenta by inducing mitochondrial ROS generation [30]. To investigate the role of oxidative stress in intestinal angiogenesis, H_2_O_2_ was employed to induce ROS generation in IPEC-J2. Here, we found that oxidative stress induced by H_2_O_2_ inhibited the proliferation and the protein expression level of VEGF-A in IPEC-J2. In addition, conditioned media from H_2_O_2_-treated IPEC-J2 was demonstrated to inhibit PVEC angiogenesis in vitro. Conclusively, these results suggested that the increased oxidative stress level in the intestine contributed to reduced intestinal angiogenesis.

DUOX2, a member of the NADPH oxidase family, is an important source of ROS [17]. In the present study, a higher expression level of DUOX2 was found in the intestine of LBW piglets. Therefore, we suspected that DUOX2 could inhibit intestinal angiogenesis by inducing excessive ROS generation in the intestine. As expected, the knockdown of DUOX2 decreased the ROS in IPEC-J2. Moreover, conditioned media from DUOX2-knockdown IPEC-J2 was demonstrated to promote PVEC angiogenesis in vitro. However, a previous study showed that DUOX2 knockdown decreases the migration and invasion of colorectal cancer cells in vitro [2]. This difference might be explained by the physiological and pathological angiogenesis induced by DUOX2. In physiological angiogenesis, DUOX2-induced ROS generation could impair the proliferation of vascular endothelial cells, thus decreasing vascular endothelial cell anagenesis. In pathological angiogenesis, DUOX2-induced ROS generation could activate proteins, such as serine/threonine kinase (AKT) and signal transducer and activator of transcription 3 (STAT3), which are known to promote tumor angiogenesis [34,35]. To explore the target of DUOX2 for angiogenesis, a real-time PCR was performed. Results showed that the mRNA expression of MMP3 was significantly up-regulated in DUOX2-knockdown IPEC-J2. Moreover, inhibiting the MMP3 could completely reverse the effect of DUOX2 knockdown on angiogenesis. Previous study has shown that MMP3 is related to angiogenesis [36]. These results suggested that increased oxidative stress and impaired intestinal anagenesis in LBW piglets might be attributed to the high expression of DUOX2.

## 5. Conclusions

Our results demonstrated the impaired angiogenesis, increased oxidative damage and DUOX2 level in jejunum for LBW piglets. DUOX2 reduces intestinal angiogenesis via ROS-MMP3 dependent mechanism. This study provided a new therapeutic target for improving the growth of LBW newborns in mammals.

## Figures and Tables

**Figure 1 antioxidants-12-01800-f001:**
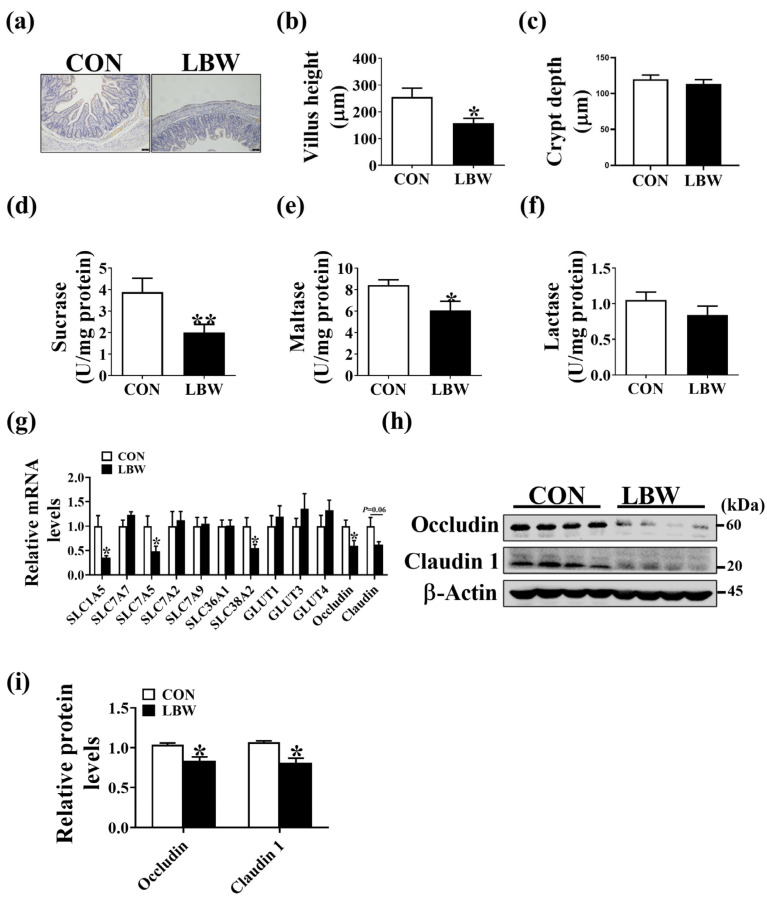
The morphology and the activities of digestive enzymes in jejunum. (**a**) Images of immunohistochemistry staining of jejunum. Bar = 75 μm. Summarized data of the intestinal villus height (**b**) and crypt depth (**c**). (**d**–**f**) The activities of sucrase, maltase, and lactase in jejunum. (**g**) The mRNA expression levels of genes related to amino acid and glucose transporters in jejunum between CON and LBW groups. (**h**,**i**) The protein levels of Occludin and Claudin 1 in the jejunum. Values are described as mean ± SEM, *n* = 10. The difference between the two groups was analyzed using the Student’s *t*-test. * *p* < 0.05, ** *p* < 0.01. CON: normal birth weight group; LBW: low birth weight group.

**Figure 2 antioxidants-12-01800-f002:**
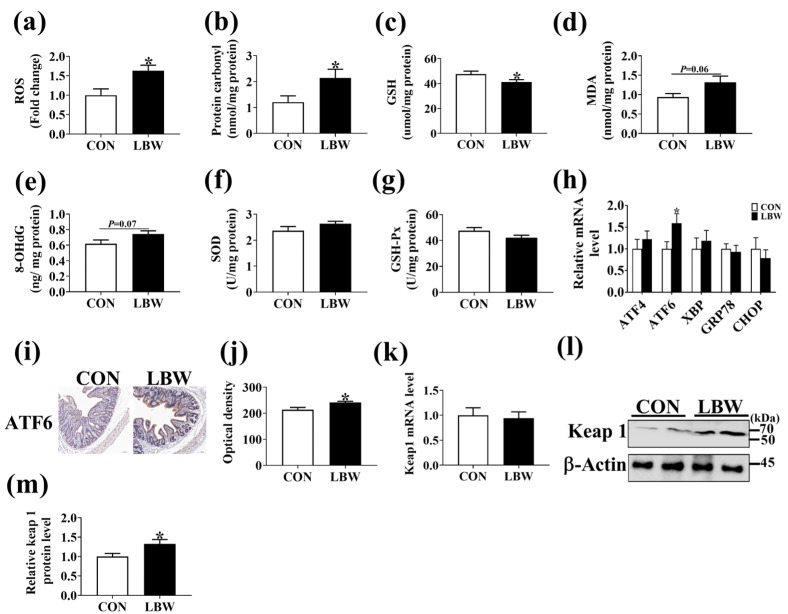
Oxidative stress level in the jejunum. The levels of ROS (**a**), protein carbonyl (**b**), GSH (**c**), MDA (**d**), 8-OHdG (**e**), SOD (**f**), and GSH-Px (**g**) in the jejunum. (**h**) The mRNA expression levels of endoplasmic stress markers. (**i**,**j**) Immunohistochemistry for ATF6 in jejunum. bar = 100 μm. (**k**) The mRNA expression level of Keap 1 in the jejunum. (**l**,**m**) The protein level of Keap1 in jejunum. Values are described as mean ± SEM, *n* = 10. CON: normal birth weight group; LBW: low birth weight group. The difference between the two groups was analyzed using the Student’s *t*-test. * indicates *p* < 0.05. CON: normal birth weight group; LBW: low birth weight group.

**Figure 3 antioxidants-12-01800-f003:**
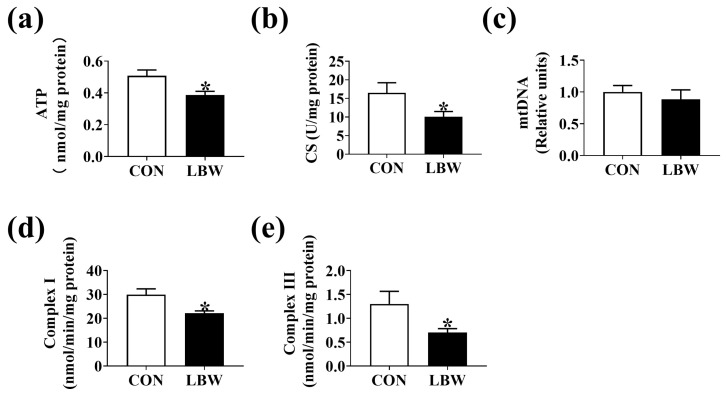
Mitochondrial biogenesis in jejunum. The levels of ATP (**a**), CS (**b**), and mtDNA (**c**) in the jejunum. The activities of Complex I (**d**) and Complex III (**e**) in the jejunum. Values are described as mean ± SEM, *n* = 10. The difference between the two groups was analyzed using the Student’s *t*-test. * indicates *p* < 0.05. CON: normal birth weight group; LBW: low birth weight group.

**Figure 4 antioxidants-12-01800-f004:**
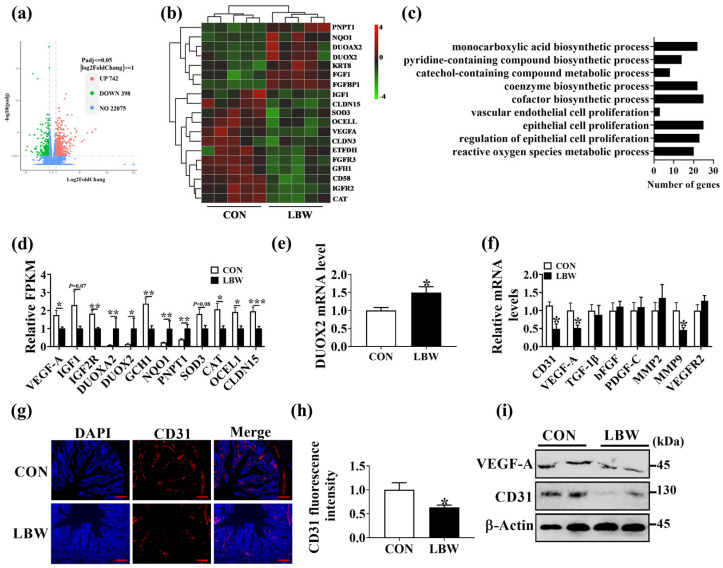
LBW increased the DUOX2 expression level and decreased vessel density in the jejunum. (**a**,**b**) Volcano map and cluster heat map illustrating the DEGs in jejunum between CON and LBW groups. *n* = 5. (**c**) GO enrichment analysis of the DEGs. *n* = 5. (**d**) The relative expression level of genes associated with angiogenesis and oxidative stress in jejunum. *n* = 5. (**e**) The mRNA expression level of DUOX2 in jejunum. (**f**) The mRNA expression levels of angiogenic factors in the jejunum. (**g**) CD31 immunofluorescence staining in jejunum. DAPI staining for the nucleus. bar = 100 μm. (**h**) Summarized data of CD31 fluorescence intensity in jejunum. (**i**) Western blotting analysis of the protein levels of VEGF-A and CD31 in jejunum. Values are described as mean ± SEM. *n* = 10. The difference between the two groups was analyzed using the Student’s *t*-test. * indicates *p* < 0.05, ** indicates *p* < 0.01, *** indicates *p* < 0.001. CON: normal birth weight group; LBW: low birth weight group.

**Figure 5 antioxidants-12-01800-f005:**
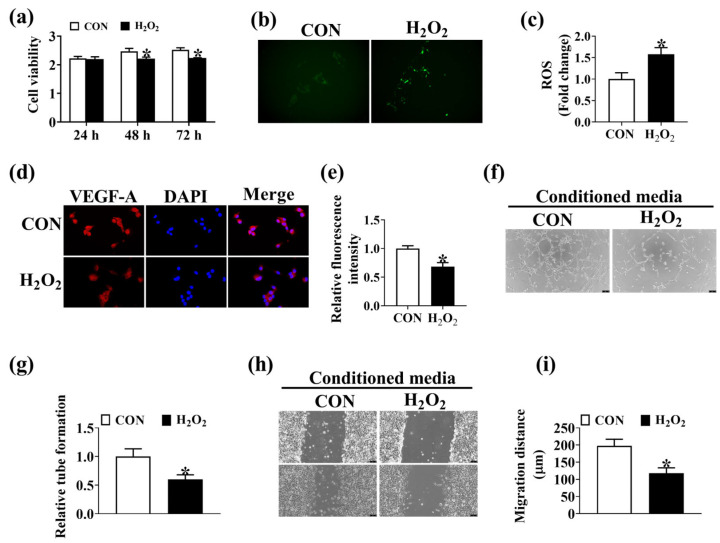
H_2_O_2_-induced oxidative stress in IPEC-J2 impairs angiogenesis of PVEC in vitro. (**a**) Proliferation assays of IPEC-J2 treated with 200 μM H_2_O_2_ for 24 h, 48 h, and 72 h. *n* = 6. (**b**,**c**) The ROS level in IPEC-J2 treated with 200 μM H_2_O_2_ for 48 h. bar = 75 μm, *n* = 6. (**d**,**e**) VEGF-A immunofluorescence staining in IPEC-J2 treated with 200 μM H_2_O_2_ for 48 h. DAPI staining for the nucleus. Bar = 200 μm, *n* = 6. Conditioned media from H_2_O_2_-treated IPEC-J2 were collected to perform PVEC tube formation (**f**,**g**) and wound healing assays (**h**,**i**). bar = 75 μm, *n* = 6. Values are described as mean ± SEM. The difference between the two groups was analyzed using the Student’s *t*-test. * indicates *p* < 0.05. All experiments were performed in triplicate.

**Figure 6 antioxidants-12-01800-f006:**
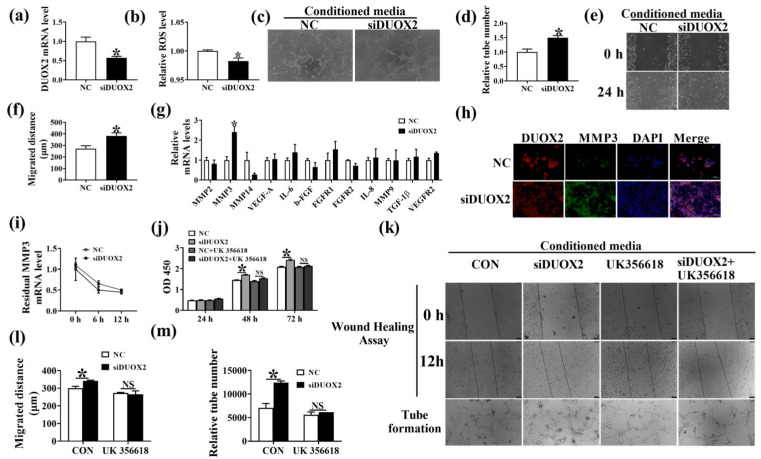
Knockdown of DUOX2 in IPEC-J2 increased PVEC angiogenesis in vitro. (**a**) The mRNA level of DUOX2 in IPEC-J2 with siRNA against DUOX2. (**b**) The ROS level in IPEC-J2 with siRNA against DUOX2. *n* = 6. Images of tube formation (**c**,**d**) and cell migration (**e**,**f**) of PVEC cultured with conditioned media from the IPEC-J2 with siRNA against DUOX2. bar = 75 μm. (**g**) The mRNA expression levels of angiogenic factors in IPEC-J2. (**h**) Images of DUOX2 and MMP3 immunofluorescence staining in IPEC-J2 with siRNA against DUOX2. bar = 200 μm. (**i**) IPEC-J2 with siRNA against DUOX2 was treated with actinomycin D (5 μg/mL) for 0, 6, and 12 h, then RNA was isolated at indicated time points. qPCR was performed to assess the mRNA expression level of MMP3. (**j**) Proliferation assay of IPEC-J2 cells. IPEC-J2 with siRNA against DUOX2 were treated with or without 10 μM MMP3 inhibitor UK356618 for 24, 48, and 72 h, respectively. Images of wound healing assay (**k**,**l**) and tube formation (**k**,**m**) of PVEC cultured with conditioned media from the NC and DUOX2 knockdown cells treated with or without 10 μM UK356618. bar = 75 μm. Values are described as mean ± SEM, *n* = 6. The difference between the two groups was analyzed using the Student’s *t*-test. * indicates *p* < 0.05. NS, no significance. All experiments were performed in triplicate.

## Data Availability

The original contributions generated for this study are included in the article, further inquiries can be directed to the corresponding author.

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
