# Peer review of "DUOX2-Induced Oxidative Stress Inhibits Intestinal Angiogenesis through MMP3 in a Low-Birth-Weight Piglet Model"

_antioxidants, 2023, doi:10.3390/antiox12101800_

Round 1

Reviewer 1 Report

Low birth weight (LBW) is associated with foetus obesity, diabetes, metabolic syndrome or deficient neurogenesis. Therefore, understanding the underlying mechanism of LBW is an interesting issue. LBW could be related with a reduced nutrient absorption through the small intestine, which would be directly connected to the villus grade of development. In this regard, blood vessels would play a central role in nutrient absorption, so in this manuscript Zou et al. suggest that a deficient try to elucidate the influence of LBW on small intestine angiogenesis by using a LBW piglet animal model, suggesting that overexpression of DUOX2 in LBW animals would induce a situation of oxidative stress and the upregulation of MMP3, which would hamper angiogenesis. Although the main conclusion of this manuscript is based on in vitro experiments, the results are interesting, and suggest a potential mechanism to explain the deficient blood vessels development in LBW animals. There are however some issues that should be addressed by the authors.

Specific comments:

1.    Line 113: please clarify what you mean by “LD muscle”

2.    Line 155: The authors claim that there is a decrease of crypt depth in the LBW group, however figure 1C does not show any difference between both groups

3.    Line 160 and figure 1h: The occluding WB image is not reliable, moreover it seems that the blot has been cut out just at the level of occluding band.

4.    Line 168: please correct the significances legends: “*P < 0.05, **P < 0.05”

5.    Lines 177-181: The authors analyse the expression of several genes, reporting an increase in ATF4, but the significance is not depicted in the corresponding figure. It is not clear in the text the reasoning for testing such genes and their relation with this investigation. If the authors want to refer to ER stress, please specify and  explain the significance of such results. The same can be applied to Keap 1 expression.

6.    Lines 192 and figure 3c: The authors report a decrease in mtDNA. However, the data are provided as “relative units”. Have the authors normalised mtDNA with respect to nuclear DNA?

7.    Lines 192-194 and figure 3. In the same line of my previous comment, a reduction of mtDNA, and the reduced activity of CS, CI and CIII could be due to a reduction of mitochondrial content in LBW. It would be very interesting to test such possibility.

8.    Lines 204. By “L and N groups” I guess that the authors refer to LBW and control groups. Please keep the same way of naming the experimental groups through the mns.

9.    Lines 210-212, Figure 4f-i. CD31 immunostaining is of low quality, it should be improved.

10. Lines 242-249. Please check and improve the redaction of this paragraph. There is a big deal of data, but it is not ease to follow, and there are some part that are inaccurate, such for instance “the promoting effects of DUOX2 on PVEC tube formation and migration in vitro”, which should be “the promoting effects of DUOX2 silencing on PVEC tube formation and migration in vitro”.

11. Perhaps more important than VEGF-A levels for angiogenesis, it would be to look at the levels of its receptor (VEGFR2). Is there any change in VEGFR2 in LBW, of upon DUOX2 silencing?.

12. Figure 5f, 6c and 6m: Tube formation assays should be confirmed by “aortic ring assays”, which is a better assay to study in vitro angiogenesis.

13.  A figure legend caption should be explicit enough to understand what is shown in the figure. In general figure legends are not fully accurate. For instance, “Figure 4. Angiogenesis in intestine. (a, b) Volcano map and cluster heat map illustrating the differentially expressed genes (DEGs)”, although it can be deduced that the authors refer to DEGs when comparing CON and LBW, it should be stated in the manuscript.

Another example Figure 6: c-f and k-m panels are described in a single sentence, when each panel should be explained individually;

(g) The mRNA expression levels in IPEC-J2C. Specify which gene or group of genes you are looking at.

Please carefully revise the accuracy of all figure legends.

14.  Is it really necessary to include the P values thought the text if they are stated in the figure legends?

 Minor editing of English language required

Author Response

Response to Reviewer 1 Comments

1. Summary

2. Questions for General Evaluation

Reviewer’s Evaluation

Response and Revisions

Does the introduction provide sufficient background and include all relevant references?

Yes

Are all the cited references relevant to the research?

Yes

Is the research design appropriate?

Yes

Are the methods adequately described?

Can be improved

We have revised the method according to your comment.

Are the results clearly presented?

Can be improved

We have revised the results according to your comment.

Are the conclusions supported by the results?

Can be improved

We have revised the results according to your comment

3. Point-by-point response to Comments and Suggestions for Authors

1.       Line 113: please clarify what you mean by “LD muscle”

Reply: We appreciate the reviewer’s comments. We revised this sentence as follows: “Total RNA was isolated from the jejunum using the TRIzol reagent.” Please see line 96.

2. Line 155: The authors claim that there is a decrease of crypt depth in the LBW group, however figure 1C does not show any difference between both groups.

Reply: We appreciate the reviewer’s comments. We revised this sentence as follows: “The results showed that a decrease of villus height was observed in the LBW group in comparison to the CON group (Figure 1a-c)” as you suggested.” Please see line 190.

3. Line 160 and figure 1h: The occluding WB image is not reliable, moreover it seems that the blot has been cut out just at the level of occluding band.

Reply: We thank the reviewer for this comment. The Occludin and β-Actin WB images were replaced according to your comment. Please see figure 1.

4. Line 168: please correct the significances legends: “*P < 0.05, **P < 0.05”

Reply: We thank the reviewer for this comment.  “*P < 0.05, **P < 0.05” has revised to “*P < 0.05, **P < 0.01” according to your comment. Please see line 209.

5.  Lines 177-181: The authors analyse the expression of several genes, reporting an increase in ATF4, but the significance is not depicted in the corresponding figure. It is not clear in the text the reasoning for testing such genes and their relation with this investigation. If the authors want to refer to ER stress, please specify and explain the significance of such results. The same can be applied to Keap 1 expression.

Reply: Many thanks for your professional comment. ROS can exacerbate protein misfolding in the ER lumen, thereby amplifying unfolding protein response (UPR) signaling (Liu et al., 2021), while inhibition of ROS level by N-acetylcysteine (NAC) could attenuate ER stress in cells (Wang et al., 2022). Keap-1/Nrf2 antioxidant defense system is an important cyto-protective mechanism against the oxidative stress associated with increased ROS and decreased antioxidant levels (Wang et al., 2017). To further confirmed that the intestine from LBW were subjected to higher oxidative stress, the mRNA expression levels of ER stress markers were determined. We added this information to the revised manuscript, please see line 216 and 350.

Liu H, Lai W, Liu X, Yang H, Fang Y, Tian L, Li K, Nie H, Zhang W, Shi Y, Bian L, Ding S, Yan J, Lin B, Xi Z. Exposure to copper oxide nanoparticles triggers oxidative stress and endoplasmic reticulum (ER)-stress induced toxicology and apoptosis in male rat liver and BRL-3A cell. J Hazard Mater. 2021, 401:123349.

Wang Y, Cui J, Zheng G, Zhao M, Hao Z, Lian H, Li Y, Wu W, Zhang X, Wang J. Ochratoxin A induces cytotoxicity through ROS-mediated endoplasmic reticulum stress pathway in human gastric epithelium cells. Toxicology. 2022, 479:153309.

Wang T, Liang X, Abeysekera IR, Iqbal U, Duan Q, Naha G, Lin L, Yao X. Activation of the Nrf2-Keap 1 pathway in short-term iodide excess in thyroid in rats. Oxid Med Cell Longev. 2017, 2017:4383652.

6. Lines 192 and figure 3c: The authors report a decrease in mtDNA. However, the data are provided as “relative units”. Have the authors normalised mtDNA with respect to nuclear DNA?

Reply: Many thanks for your professional comment. mtDNA was normalized to the acidic ribosomal phosphoprotein P0 (36B4) nuclear gene. We added this information to the revised manuscript, please see line 93.

7. Lines 192-194 and figure 3. In the same line of my previous comment, a reduction of mtDNA, and the reduced activity of CS, CI and CIII could be due to a reduction of mitochondrial content in LBW. It would be very interesting to test such possibility.

Reply: Many thanks for your professional comment. Fresh intestinal samples are needed for detection of mitochondrial content in intestine. However, the intestinal sample were frozen by liquid nitrogen.

8. Lines 204. By “L and N groups” I guess that the authors refer to LBW and control groups. Please keep the same way of naming the experimental groups through the mns.

Reply: We thank the reviewer for this comment.  “L and N groups” has revised to “LBW and CON groups” as you suggested.

9. Lines 210-212, Figure 4f-i. CD31 immunostaining is of low quality, it should be improved.

Reply: We thank the reviewer for this comment. We revised it as you suggested, please see figure 4g

10. Lines 242-249. Please check and improve the redaction of this paragraph. There is a big deal of data, but it is not ease to follow, and there are some part that are inaccurate, such for instance “the promoting effects of DUOX2 on PVEC tube formation and migration in vitro”, which should be “the promoting effects of DUOX2 silencing on PVEC tube formation and migration in vitro”.

Reply: Thank you for your valuable comments and suggestions on our manuscript.   We revised this paragraph as you suggested, please see line 306.

11. Perhaps more important than VEGF-A levels for angiogenesis, it would be to look at the levels of its receptor (VEGFR2). Is there any change in VEGFR2 in LBW, of upon DUOX2 silencing?

Reply: Many thanks for your professional comment. The result of mRNA level of VEGFR2 was presented in Figure 4 and 6 according to your comment, please see the revised manuscript. The results showed that no difference was observed in the mRNA expression level of VEGFR2 between LBW and NBW group. DUOX2 silencing showed no effect on the mRNA expression level of VEGFR2.

12. Figure 5f, 6c and 6m: Tube formation assays should be confirmed by “aortic ring assays”, which is a better assay to study in vitro angiogenesis.

Reply: We appreciate the reviewer’s comments. Aortic ring assays is a better assay to study angiogenesis in vitro. However, this assay is new for us, it is difficult for us to complete this assay in a short time. We realized the weakness in our current work according to your comment, and we will improve our scientific research level in the future work.

13. A figure legend caption should be explicit enough to understand what is shown in the figure. In general figure legends are not fully accurate. For instance, “Figure 14. Angiogenesis in intestine. (a, b) Volcano map and cluster heat map illustrating the differentially expressed genes (DEGs)”, although it can be deduced that the authors refer to DEGs when comparing CON and LBW, it should be stated in the manuscript.

Reply: We appreciate the reviewer’s comments. We have revised the figure legend according to your comments. Please see the revised manuscript.

Another example Figure 6: c-f and k-m panels are described in a single sentence, when each panel should be explained individually;

Reply: Thanks for your professional comment. We have revised this according to your comment. Please see line 314 and 322.

(g) The mRNA expression levels in IPEC-J2C. Specify which gene or group of genes you are looking at.

Reply: Thank you for your valuable comments. We revised it as follows: “(g) The mRNA expression levels of angiogenic factors in IPEC-J2” as you suggested, please see line 316.

Please carefully revise the accuracy of all figure legends.

Reply: We appreciate the reviewer’s comments. We have revised the figure legends according to your comment. Please see the revised manuscript.

14.  Is it really necessary to include the P values thought the text if they are stated in the figure legends?

Reply: Thank you for your valuable comments. The P values was removed according to your comment. Please see the revised manuscript.

4. Response to Comments on the Quality of English Language

1: Minor editing of English language required

Reply: We appreciate the reviewer’s comments. We have revised the manuscript as you suggested.

Reviewer 2 Report

The study entitled: „DUOX2-induced oxidative stress inhibits intestinal angiogenesis through MMP3 in a low-birth-weight piglet model” is interesting and relatively well conducted. The spectrum of method is very broad.

However, the study suffers from several flaws, inconsequence and errors that need to be addressed. On many occasions the rationale for evaluating expressions or activities of particular molecules is not provided. This leads to a impression of chaotic jumping from one aspect to another and thus the study lacks some complementarity and complexity.

Major:

What is the rationale for evaluating the expression of AA transporters (SLC1A5, SLC7A5, and SLC38A2) and at the same time activity maltase and sucrase and protein expression of Occludin. This seems to be very chaotic.

Figure 1 legend is very superficial as it states i.e for F1g The mRNA expression levels of genes in intestine.

In my opinion glucose transporters should be evaluated in parallel on both mRNA and protein leveles, AA transporters on protein level and Occludin on  mRNA level in order to make it more comprehensive and complete.

What is meant by this statement? „a trend of the increased level of MDA” – was it trend in time?

And similarly for Figure 2 why IHC was performed only for ATF6 and not the others? Why there is no Keap1 mRNA expression performed?

Are all the changes in expression levels between LBW and CON reflected in transcriptome analysis? i.e CD31 mRNA expression?

What about DUOX2 on protein levels in LBW and CON?

Minor clerical errors

Author Response

Response to Reviewer 2 Comments

1. Summary

2. Questions for General Evaluation

Reviewer’s Evaluation

Response and Revisions

Does the introduction provide sufficient background and include all relevant references?

Yes

Are all the cited references relevant to the research?

Yes

Is the research design appropriate?

Must be improved

We added more information about the research design to the manuscript.

Are the methods adequately described?

Can be improved

We have revised the method according to your comment.

Are the results clearly presented?

Must be improved

We have revised the results according to your comment.

Are the conclusions supported by the results?

We have revised the results according to your comment

3. Point-by-point response to Comments and Suggestions for Authors

1. The study entitled: DUOX2-induced oxidative stress inhibits intestinal angiogenesis through MMP3 in a low-birth-weight piglet model” is interesting and relatively well conducted. The spectrum of method is very broad.

Reply: Thank you for your valuable comments. We revised the method according to your comment. Please see the section of “Materials and Methods”

2. However, the study suffers from several flaws, inconsequence and errors that need to be addressed. On many occasions the rationale for evaluating expressions or activities of particular molecules is not provided. This leads to a impression of chaotic jumping from one aspect to another and thus the study lacks some complementarity and complexity.

Reply: Thank you for your valuable comments. We revised the manuscript according to your comment, please see the revised manuscript.

Major:

3. What is the rationale for evaluating the expression of AA transporters (SLC1A5, SLC7A5, and SLC38A2) and at the same time activity maltase and sucrase and protein expression of Occludin. This seems to be very chaotic.

Reply: Many thanks for your professional comment. The sentence of “Low birth weight (LBW) was reported to impair the nutrient digestion and absorption, as well as the protein expression of Occludin in intestine of piglets [17,18]. To further confirm the successful establishment of the LBW model, these indexes were evaluated.” was added to the manuscript according to your comments. Please see line 187.

4. Figure 1 legend is very superficial as it states i.e for F1g The mRNA expression levels of genes in intestine.

Reply: Thank you for your valuable comments. We revised it as follows: “(g) The mRNA expression levels of genes related to amino acid and glucose transporters in intestine between CON and LBW groups.” as you suggested. Please see line 205.

5. In my opinion glucose transporters should be evaluated in parallel on both mRNA and protein leveles, AA transporters on protein level and Occludin on mRNA level in order to make it more comprehensive and complete.

Reply: We appreciate the reviewer’s comments. The results of mRNA levels of glucose transporters and Occludin, and the protein expression level of Claudin 1 were added to the revised manuscript as you suggested, please see figure 1.

6. What is meant by this statement? „a trend of the increased level of MDA” – was it trend in time?

Reply: Thank you for your valuable comments. We revised this sentence as follows: “Compared with the CON group, the LBW group has a tendency to increase the concentrations of MDA and 8-OHdG in jejunum (Figure 2d, e).” Please see line 214.

7. And similarly for Figure 2 why IHC was performed only for ATF6 and not the others?

Reply:Thank you for your valuable comments. We found that the mRNA level of ATF6 was higher in the LBW group in comparison to the CON group, however, no significant difference was found in the mRNA expression level of ATF4, XBP, GRP78, or CHOP between the two group. Therefore, IHC was performed to further investigate the expression level of ATF6 in jejunum.

8. Why there is no Keap1 mRNA expression performed?

Reply:Thank you for your valuable comments. The result of Keap1 mRNA expression level was added to the manuscript according to your comment. Please see figure 2k.

9. Are all the changes in expression levels between LBW and CON reflected in transcriptome analysis? i.e CD31 mRNA expression?

Reply: Many thanks for your professional comment. A total of 1140 differentially expressed genes (DEGs) were identified in jejunum. In the present study, the DEGs related to oxidative stress and angiogenesis was showed in the supplemental table 2. The mRNA expression of CD31 was lower in LBW group compared to CON group. Please see figure 4f.

10. What about DUOX2 on protein levels in LBW and CON?

Reply: We appreciate the reviewer’s comments. We have purchased several DUOX2 primary antibodies from different companies (Abcam, Novus Biologicals). But due to the species specificity, we did not get the results of DUOX2 protein level in jejunum of piglets. We only obtained the result of mRNA level of DUOX2 between the two groups. We found the mRNA level of DUOX2 was higher in LBW group compared to CON group. This result was added to the manuscript according to your comment. Please see figure 4e.

4. Response to Comments on the Quality of English Language

1: Minor clerical errors

Reply: We appreciate the reviewer’s comments. We have revised the manuscript as you suggested.

Reviewer 3 Report

The article entitled “DUOX2-induced oxidative stress inhibits intestinal angiogenesis through MMP3 in a low-birth-weight piglet model” aims to analyze the mechanisms underlying low birth weight (LBW).

The in vivo results show that in accordance with the reduced birth weight the villus height and the activity of maltase in intestine were lower in Low compared to the control weight group. Furthermore, the low weight group exhibited a higher oxidative stress level and impaired mitochondrial function in intestine, and was lower than CON group in the intestinal vascular density.

To investigate the potential link between intestinal cell oxidative stress and vascular density, the authors perform in vitro studies of indirect co-culture between intestinal cells and endothelial cells. The in vitro results confirm the in vivo data that Conditioned media (CM) from porcine intestinal epithelial cells (IPEC-J2) with H2O2 treatment decreased the angiogenesis of porcine vascular endothelial cells (PVEC).

To analyze the mechanism of action, the authors analyze differentially expressed genes (DEGs) in intestine between Low and Normal groups. DEG results show A higher expression level of dual oxidase 2 (DUOX2) was found in intestine for LBW piglets.

Following experiment aimed to verify the mechanistic role of duox2. Results showed that Knockdown of DUOX2 in intestinal IPEC-J2 increased the proliferation and decreased the oxidative stress level. Conditioned medium from IPEC-J2 with DUOX2-knockdown promoted angiogenesis of PVEC. Mechanistically, knockdown of DUOX2 decreased reactive oxygen species (ROS) level, thus increasing the angiogenesis in a matrix metalloproteinase 3 (MMP3) dependent manner.

The authors conclude that

DUOX2-induced oxidative stress inhibited intestinal angiogenesis through MMP3 in a LBW piglet model.

In my opinion the major concern concerns the experiments underlying the analysis of the DEGs. The authors base the article on DEG data between normal weight and low weight pigs. However, these experiments are unclear and poorly described. The authors are requested to describe more fully the method used and the statistic used on the basis of which the most modified DEGs are DUOX and VEGF, please attach the original data as an excel file in the supplementary materials.

the authors should better introduce the two groups used for the study and always use the same denomination (e.g. Low and Normal and not L or N or LBW or Con

In the abstract and in the text, explain why the in vitro system is used.

For a better reading of the manuscript it is advisable to reduce the non-useful acronyms such as d es CM or in “Total RNA was isolated from the LD muscle using the TRIzol reagent “. What' LD? Please write in full.

Explain more about the role of MMP3 and why MMP3 inhibitor is used

What happens in the co-culture of intestinal cells and endothelial cells? What is the activity of NADPH oxidase in the two cell types and what is the expression of the catalytic and regulatory subunits of NADPH oxidase?

Fig. 1g: Please increase the size of the panel to make it easier to read. It would be interesting to see the protein expression of all genes, and in particular of claudin whose significance is borderline as a messenger but could be significant as a protein.

Paragraph 3.5: Please pintroduce the experiments carried out and the reason why they were carried out. 

Fig. 4: Plese Increase the size of panels c and d. They are not legible

Author Response

Response to Reviewer 3 Comments

1. Summary

2. Questions for General Evaluation

Reviewer’s Evaluation

Response and Revisions

Does the introduction provide sufficient background and include all relevant references?

Can be improved

We have revised the introduction according to your comment.

Are all the cited references relevant to the research?

Can be improved

We have revised the references according to your comment.

Is the research design appropriate?

Can be improved

We added more information about the research design to the manuscript according to your comment.

Are the methods adequately described?

Must be improved

We have revised the method according to your comment.

Are the results clearly presented?

Can be improved

We have revised the results according to your comment.

Are the conclusions supported by the results?

Can be improved

We have revised the conclusions according to your comment

3. Point-by-point response to Comments and Suggestions for Authors

1. The article entitled “DUOX2-induced oxidative stress inhibits intestinal angiogenesis through MMP3 in a low-birth-weight piglet model” aims to analyze the mechanisms underlying low birth weight (LBW).

The in vivo results show that in accordance with the reduced birth weight the villus height and the activity of maltase in intestine were lower in Low compared to the control weight group. Furthermore, the low weight group exhibited a higher oxidative stress level and impaired mitochondrial function in intestine, and was lower than CON group in the intestinal vascular density.

To investigate the potential link between intestinal cell oxidative stress and vascular density, the authors perform in vitro studies of indirect co-culture between intestinal cells and endothelial cells. The in vitro results confirm the in vivo data that Conditioned media (CM) from porcine intestinal epithelial cells (IPEC-J2) with H2O2 treatment decreased the angiogenesis of porcine vascular endothelial cells (PVEC).

To analyze the mechanism of action, the authors analyze differentially expressed genes (DEGs) in intestine between Low and Normal groups. DEG results show A higher expression level of dual oxidase 2 (DUOX2) was found in intestine for LBW piglets.

Following experiment aimed to verify the mechanistic role of duox2. Results showed that Knockdown of DUOX2 in intestinal IPEC-J2 increased the proliferation and decreased the oxidative stress level. Conditioned medium from IPEC-J2 with DUOX2-knockdown promoted angiogenesis of PVEC. Mechanistically, knockdown of DUOX2 decreased reactive oxygen species (ROS) level, thus increasing the angiogenesis in a matrix metalloproteinase 3 (MMP3) dependent manner.

The authors conclude that

DUOX2-induced oxidative stress inhibited intestinal angiogenesis through MMP3 in a LBW piglet model.

In my opinion the major concern concerns the experiments underlying the analysis of the DEGs. The authors base the article on DEG data between normal weight and low weight pigs. However, these experiments are unclear and poorly described. The authors are requested to describe more fully the method used and the statistic used on the basis of which the most modified DEGs are DUOX and VEGF, please attach the original data as an excel file in the supplementary materials.

Reply: We appreciate the reviewer’s comments. We revised the section of “Materials and Method” according to your comment. Please see the revised manuscript.

We attached the original data in the supplementary materials as you suggested. Please see supplemental Table 2.

2. the authors should better introduce the two groups used for the study and always use the same denomination (e.g. Low and Normal and not L or N or LBW or Con

Reply: We thank the reviewer for this comment.  “L and N groups” has revised to “LBW and CON groups” as you suggested.

3. In the abstract and in the text, explain why the in vitro system is used.

Reply: Thanks for your comment. The information was added to the abstract and text as you suggested, please see line 14, 279, and 297.

4. For a better reading of the manuscript it is advisable to reduce the non-useful acronyms such as d es CM or in “Total RNA was isolated from the LD muscle using the TRIzol reagent “. What' LD? Please write in full.

Reply: We appreciate the reviewer’s comments. “CM” was deleted according to your comment. Please see the revised manuscript.

We revised this sentence as follows: “Total RNA was isolated from the jejunum using the TRIzol reagent.” Please see line 96.

5. Explain more about the role of MMP3 and why MMP3 inhibitor is used

Reply: Many thanks for your professional comment. DUOX2 knockdown in IPEC-J2 promote MMP3 expression. MMP3 is a proangiogenic factor, we questioned whether MMP3-mediated angiogenesis promotion is controlled by DUOX2 knockdown. Therefore, MMP3 inhibitor was employed to inhibit MMP3 activity.

6. What happens in the co-culture of intestinal cells and endothelial cells? What is the activity of NADPH oxidase in the two cell types and what is the expression of the catalytic and regulatory subunits of NADPH oxidase?

Reply: We appreciate the reviewer’s comments. In the present study, intestinal cells with DUOX2 knockdown can secrete angiogenic factors (e.g., MMP3) into the medium, thereby affecting vascular endothelial cells proliferation, tube formation, and migration.

Based on the results of transcriptome analysis, no significant differences were observed in the NOX1, NOX2, NOX3, NOX4, NOX5, or DUOX1 in the jejunum between CON and LBW groups (supplemental table 2). Therefore, we investigated the role of DUOX2 in intestinal angiogenesis. DUOX2 is expressed in the tip of epithelial cells in the intestine and require maturation factor proteins DUOXA2 (Grasberger et al., 2006). In the present study, the expression level of DUOX2A2 was upregulated in the LBW group compared to CON group (Figure 4d).

Grasberger, H. Refetoff, S. Identification of the maturation factor for dual oxidase. Evolution of an eukaryotic operon equivalent. J. Biol. Chem. 2006, 18269–18272.

Sommer, F. Backhed, F. The gut microbiota engages different signaling pathways to induce Duox2 expression in the ileum and colon epithelium. Mucosal Immunol. 2015, 372–379.

7. Fig. 1g: Please increase the size of the panel to make it easier to read. It would be interesting to see the protein expression of all genes, and in particular of claudin whose significance is borderline as a messenger but could be significant as a protein.

Reply: Many thanks for your comment. We have revised Fig.1g according to your comment. The result of protein level of claudin was presented in Figure 1 as you suggested.

8. Paragraph 3.5: Please introduce the experiments carried out and the reason why they were carried out. 

Reply: We appreciate the reviewer’s comments. The results (Figure 2, 3) indicated that LBW exhibited a higher oxidative stress level and a lower intestinal vascular density in jejunum. Oxidative stress plays a crucial role in angiogenesis, which could promote to the aberrant angiogenesis by inducing dysfunction of vascular endothelial cells (Huang et al., 2019). Based on these results, we hypothesis that the reduced intestinal angiogenesis could be due to an elevation of oxidative stress in LBW intestine. Therefore, this experiment was carried out to test this hypothesis.

Huang, Y. J, Nan, G. X. Oxidative stress-induced angiogenesis. J. Clin. Neurosci. 2019, 63, 13-16

9. Fig. 4: Plese Increase the size of panels c and d. They are not legible

Reply: Many thanks for your comment. We revised figure 4 as you suggested, please see figure 4.

Round 2

Reviewer 1 Report

The authors have responded satisfactorily to all the comments raised.

Author Response

The authors have responded satisfactorily to all the comments raised.

Reply: We appreciate the reviewer’s comments. 

Reviewer 2 Report

Generally the improvements to the manuscript are acceptable. However, there are some clerical errors in gene symbols and wrong use of italics throughout the text i.e. Line 197 “level of glucose transporters GLUT1, GlUT3, or GLUT4 (Figure 1g).”

It might be very useful to add this comment (or adequately rephrased) on DUOX2 Western blotting as depicted by the authors: “We appreciate the reviewer’s comments. We have purchased several DUOX2 primary antibodies from different companies (Abcam, Novus Biologicals). But due to the species specificity, we did not get the results of DUOX2 protein level in jejunum of piglets. We only obtained the result of mRNA level of DUOX2 between the two groups. “  in i.e. Figure 4 legend or in the Materials and Methods section.

Author Response

Response to Reviewer 2 Comments

1. Summary

2. Questions for General Evaluation

Reviewer’s Evaluation

Response and Revisions

Does the introduction provide sufficient background and include all relevant references?

Yes

Are all the cited references relevant to the research?

Yes

Is the research design appropriate?

Yes

Are the methods adequately described?

Yes

Are the results clearly presented?

Yes

Are the conclusions supported by the results?

Yes

3. Point-by-point response to Comments and Suggestions for Authors

1. Generally the improvements to the manuscript are acceptable. However, there are some clerical errors in gene symbols and wrong use of italics throughout the text i.e. Line 197 “level of glucose transporters GLUT1, GlUT3, or GLUT4 (Figure 1g).”

Reply: We appreciate the reviewer’s comments. We have revised the gene symbols and italics according to your comments. Please see line 17, 200, 222, 224, 256, 261, 275, 316, and 323.

We revised this sentence as follows: “However, no difference was observed in the mRNA expression level of glucose transporters GLUT1, GLUT3, or GLUT4 between the two groups (Figure 1g).”

2. It might be very useful to add this comment (or adequately rephrased) on DUOX2 Western blotting as depicted by the authors: “We appreciate the reviewer’s comments. We have purchased several DUOX2 primary antibodies from different companies (Abcam, Novus Biologicals). But due to the species specificity, we did not get the results of DUOX2 protein level in jejunum of piglets. We only obtained the result of mRNA level of DUOX2 between the two groups. “  in i.e. Figure 4 legend or in the Materials and Methods section.

Reply: We appreciate the reviewer’s comments. The sentence of “Several DUOX2 primary antibodies were purchased from different companies (e.g., Abcam and Novus Biologicals). But due to the species specificity, the result of DUOX2 protein level in jejunum of piglets was not obtained. The mRNA level of DUOX2 between the CON and LBW groups was determined by qPCR.” was added to the section of Materials and Method” as you suggested. Please see line 114. 

4. Response to Comments on the Quality of English Language

5. Additional clarifications

Reviewer 3 Report

The authors responded to my comments. I have no other requests. 

Author Response

The authors responded to my comments. I have no other requests. 

Reply: We appreciate the reviewer’s comments. 
